# Group structure and individual relationships of sanctuary-living Grauer's gorillas (*Gorilla beringei graueri*)

**Austin Leeds**[1]*, **Dalmas Kakule**[2], **Laura Stalter**[1], **Jackson K. Mbeke**[2], **Katie Fawcett**[2]

**1** Animals, Science and Environment, Disney's Animal Kingdom®, Lake Buena Vista, Florida, United States of America, **2** Gorilla Rehabilitation and Conservation Education Center, Kasugho, North Kivu, Democratic Republic of the Congo

* austin.leeds@disney.com

**Data Availability Statement:** All relevant data are within the paper and its Supporting information files.

## Abstract

The study of individual social relationships and group structure provides insights into a species' natural history and can inform management decisions for animals living in human care. The Gorilla Rehabilitation and Conservation Education (GRACE) center provides permanent sanctuary for a group of 14 Grauer's gorillas (*Gorilla beringei graueri*), a critically endangered and poorly studied subspecies of the genus gorilla, in the Democratic Republic of the Congo. We monitored the association patterns of the gorillas at GRACE over eight months and here describe their individual relationships and group structure via multiple social network statistics. The group was highly connected but associations between individuals were weak on average. Social network metrics describe that an adult female was the most gregarious and socially central individual within the group. In fact, adult females were the most gregarious and socially central on average. Group level association patterns were significantly correlated over the study period and across observation types, suggesting the group was socially stable during the eight month study period. The data collected in this study were done so by GRACE caregivers as part of their daily husbandry routine and provided important insights into this group's behavior, ultimately informing on their care, welfare and future release considerations. The methodological approaches implemented here are easily scalable to any primate sanctuary or care facility seeking to use data to inform husbandry and management procedures. Lastly, our study is the first social network analysis to be conducted on Grauer's gorillas and provides tentative insights into the behavior of this poorly studied subspecies. Though more research is needed to evaluate if the findings here are reflective of this subspecies' natural history or the idiosyncrasies of the group.

## Introduction

The study of primate social structure considers how individual relationships, that is the content, quality and patterning of interactions between individuals, contribute to groupings as a whole [1]. From a basic perspective this field of study has provided insights into how broad group demographics such as group size [2, 3], sex ratios [4] and kinship [5–8], individual

**Funding:** The authors received no specific funding for this work.

**Competing interests:** The authors have declared that no competing interests exist.

benefits such as fitness outcomes [9–12] and stress mitigation [13–15] and resource availability [16–18] shape primate societies. This line of research has expanded to additionally encompass more applied perspectives. For example, how social structures vary in response to anthropogenic disturbances, thus informing conservation practices [19–21] and how social structures take shape in human care, ultimately informing species management and individual care and welfare [22, 23].

The genus gorilla is divided into two species (western, *Gorilla gorilla*; eastern, *Gorilla beringei*), of which two subspecies have been the focus of intense field study for over five (mountain gorillas, *G. b. beringei*) and three (western lowland gorillas, *G. g. gorilla*) decades, respectively [24, 25], and have provided the foundation for our understanding of gorilla social structure. Male-female relationships are considered the core relationship of gorilla groups [24]. In single-male mountain and western lowland gorilla groups, females spend more time associated with the silverback than with other females and maturing males [26, 27], and in multi-male mountain gorilla groups, females tend to associate with the highest ranking male over lower ranking males [28]. Interestingly, despite the heavy emphasis on male-female relationships, male-immature mountain gorilla dyads have stronger association patterns than male-female dyads, highlighting that immature gorillas are an integral component of gorilla social structure [4]. Preferential associations have been observed amongst females, however, these relationships do not appear to last longer than two years, suggesting females do not form strong long-term relationships with other adult females [29].

Grauer's gorillas (*G. b. graueri*) are a critically endangered subspecies of eastern gorilla endemic to the Democratic Republic of the Congo (DRC) [30]. Though Grauer's gorillas have been studied since the 1970's [31, 32] significantly less has been reported on their behavior, particularly their social structure, compared to western and mountain gorillas [33]. Grauer's gorillas live in mixed-sex groups similar in size to mountain and western lowland gorillas [24, 34] that are predominantly single-male [35]. Females with infants have been observed to spend more time resting together collectively, while females without offspring did not rest together with the same frequency, though this may be biased by a preference for females with offspring to maintain associations with the silverback [32]. Few data on male-male dynamics are available, though evidence suggests blackback males appear to avoid silverbacks within their natal group [32].

In response to a growing population of orphaned Grauer's gorillas living in human care, the Gorilla Rehabilitation and Conservation Education Center (GRACE) was opened in 2010 in the North Kivu Province, DRC, to provide permanent sanctuary for rescued Grauer's gorillas. Currently GRACE is home to a single group of 14 gorillas (Table 1), including an adult male silverback and two maturing blackback males, all of whom are presumed to be unrelated. The gorillas living at GRACE are the only individuals of this subspecies in the world living in human care and are considered candidates for release. In support of GRACE's operation and mission, we implemented an ongoing behavior monitoring program focused on the association patterns amongst the individual gorillas. This methodology was selected because associations are regularly used to define social relationships of gorillas in nature [4, 29, 36, 37] and could be readily implemented by the animal care team as part of their day-to-day routine.

This monitoring program supports GRACE's operation and mission in three primary ways. First, we seek to ensure the gorillas at GRACE have optimal care and welfare. Social relationships are significant contributors to the welfare status of primates [22, 38–40] and are sensitive to a variety of factors in human care. For example, primate social relationships can change in response to the introduction of new conspecifics [41–43], space availability [44, 45], participation in cognitive research [46, 47], the early life history of group members [48] and can also simply vary with time and developmental status of individuals [22, 49, 50]. Thus monitoring

**Table 1. Gorilla identities and demographic information.**

| Identity | Sex | Developmental Status | Estimated Birth Year | Estimated Age in 2022 | Arrival Year | Origin | Fertility Status* |
|----------|-----|---------------------|---------------------|----------------------|--------------|--------|-------------------|
| AMA | F | Adult | 2008 | 14 | 2010 | Goma | On contraceptive |
| ISA | F | Adult | 2012 | 10 | 2013 | Single | On contraceptive |
| ITE | F | Adult | 2003 | 19 | 2011 | Kinigi | On contraceptive |
| KAL | F | Adult | 2013 | 9 | 2015 | Single | On contraceptive |
| KIG | M | Silverback | 2008 | 14 | 2010 | Goma | Intact |
| LUB | M | Blackback | 2009 | 13 | 2011 | Single | Intact |
| LUL | F | Subadult | 2015 | 7 | 2016 | Single | On contraceptive |
| MAP | F | Adult | 2004 | 18 | 2010 | Goma | On contraceptive |
| MUY | F | Adult | 2011 | 11 | 2014 | Single | On contraceptive |
| NDJ | F | Adult | 2009 | 13 | 2010 | Goma | On contraceptive |
| PIN | F | Adult | 2001 | 21 | 2011 | Kinigi | On contraceptive |
| SER | F | Adult | 2002 | 20 | 2011 | Kinigi | On contraceptive |
| SHA | M | Blackback | 2010 | 12 | 2011 | Single | Intact |
| TUM | F | Adult | 2006 | 16 | 2011 | Kinigi | On contraceptive |

*All females receive oral estrogen/progestin combination contraceptive pill.

and quantifying social relationships within groups is an important procedure for ensuring optimal primate welfare [38, 51]. We actively utilize these data at GRACE to assess the quality of individual relationships and group cohesion and to make informed management decisions. Second we plan to use these data to inform future release efforts. For example, identifying individuals who maintain strong social bonds and thus may support each other during the stresses of a release may contribute to release success. Similarly beneficial could be the identification of individuals who are particularly social with many different gorillas at GRACE and thus may more readily socially integrate with gorillas in nature. Lastly, we hope these data may add to our understanding of this poorly studied subspecies' behavior more broadly. Though the GRACE group lives in human care they are managed in social and ecological contexts that mimic natural settings and thus may provide tentative insights into Grauer's gorilla behavior, similar to a growing body of primate behavior research being conducted within in-situ sanctuaries [for discussion and review see 52, 53].

Here we present data collected during the first eight months of this long-term monitoring program. We monitored association frequencies in three specific contexts: during their morning forage in a large outdoor yard, shifting from their forest enclosure to their night houses, and while nesting in their night houses. These three contexts were evaluated because they allowed for regular monitoring of the gorillas by care staff who were the primary data collectors for this project and each context was an important component of the gorilla's daily lives, potentially informing on how associations are influenced by management practices and general group activity. To this later point, mountain gorilla social structure can vary based on group activity [54], thus by evaluating the GRACE group across contexts a more complete analysis of social structure can be determined. In this study we specifically sought to:

1. Describe association indices at the group level and by gorilla origin and sex.

2. Calculate social network statistics for each individual and describe patterns by gorilla origin, sex and age.

3. Test consistency of association indices across time to evaluate group stability and by observation context to evaluate how management/group activity affects group structure.

4. Test group structure for the presence of "hyper-social" members and subgroupings.

Within the above analyses we placed additional focus on the relationships of the three males within the group as this subspecies predominantly lives in single male groups in nature [35]. We considered this analysis exploratory and thus adopted no a priori hypotheses because this is the first detailed analysis of this group's social structure and one of the few studies of Grauer's gorillas in general.

## Methods

### Ethical note

This study was observational, non-invasive and completed by professional caregivers providing daily care to the gorillas. This study complied with the American Society of Primatologists principles for the ethical treatment of non-human primates. The GRACE sanctuary is accredited by the Global Federation of Animal Sanctuaries.

### Study group

We studied a group of 14 ($n_{male}$ = 3; $n_{female}$ = 11) Grauer's gorillas living at the GRACE sanctuary (Table 1). Each gorilla arrived at GRACE via a confiscation/rescue, and thus the age of each individual is an estimate. Six gorillas arrived at the GRACE facility as single confiscations/rescue events, four arrived as a group from a rescue facility in Goma, DRC, and four arrived as a group from a rescue facility in Kinigi, Rwanda. The Goma and Kinigi gorillas were originally single confiscations/rescue events that were then housed together for socialization prior to the construction of GRACE. It is presumed that none of the gorillas are related. Once arriving at GRACE, gorillas were integrated into the group at rates specific to their recovery needs. The study group has been integrated as a single social group since 2016, following the arrival and integration of the most recent rescued individual.

Gorillas at GRACE are housed in six different areas: two interconnected night houses composed of 10 individual rooms totaling approximately 210 $m^2$ in size, an outdoor yard (mixing yard) approximately 3,000 $m^2$ in size and two large forested enclosures approximately 81,600 $m^2$ and 46,300 $m^2$ in size. In a typical 24 hr period the gorillas spend the night in the night houses, choosing which room and which gorillas to be with. In the morning the gorillas are then moved to the mixing yard where they have access to provided diet items for approximately one hour. Provided dietary items include vegetation harvested from the forest enclosures (e.g. elephant grass, Rubus sp.) and fruit purchased from local markets (e.g. passion fruit, banana, pineapple, plum), all of which are spread throughout the mixing yard. The group then shifts to one of the two forest enclosures for the day. Forest enclosure access is provided in blocks of time such that gorillas have access to one enclosure for approximately six to nine months then have access to the alternative enclosure for a similar amount of time. During this study the gorillas had access to the smaller forest enclosure each day. Vegetation grows freely throughout both enclosures that the gorillas are free to forage on. Midday, the gorillas are recalled into the mixing yard, followed by a return to the forest enclosure. At the end of the day the gorillas are recalled back to the mixing yard and then to the night houses.

### Data collection

Data were collected over 32 weeks beginning in early May 2022 through mid-December 2022 in three contexts: nesting ($n_{observations}$ = 192), mixing yard ($n_{observations}$ = 157) and forest enclosure shifting ($n_{observations}$ = 159). For each context, each gorilla dyad (91 total) was recorded as being associated (within 1 m) or not. Studies of gorilla associations in nature frequently utilize

within 5m as their metric [27, 29], however, recent research has highlighted closer metrics (within 2m) provide more definition in association measures [4]. We thus sought to adopt a similarly conservative approach, however, we utilized 1m as it provided a more reliably trained metric as it was approximately equal to an adult gorilla arm length. This visual cue aided researchers who at times observed the gorillas from distances greater than 25m.

For nesting observations, data collectors entered the night house upon arrival to GRACE in the morning and conducted a single group scan in which each pair of gorillas nesting in association was recorded. Typically, all gorillas were still in their nests upon observer arrival. If one gorilla was out of a nest and their nest could be identified, their associations were recorded based on nest location. If multiple gorillas were out of their nests, they were marked as not visible for that observation. For mixing yard observations, data were collected in the morning after the gorillas were shifted from the night house to the mixing yard via three group scans. The first scan occurred approximately 15 min following the release of the gorillas into the yard. The second and third scan occurred approximately 15 and 30 min later, respectively. For each scan, each associated pair of gorillas was recorded. Forest enclosure shifting observations occurred in the late afternoon. Typically, the gorillas congregate near the shift door from the forest enclosure to the mixing yard at the end of the day prior to their recall. Approximately 15 min prior to husbandry staff signaling for the recall, a single group scan of the gorillas in the forest enclosure occurred where each associated pair was recorded.

All data collectors were caregivers at GRACE and could reliably identify each individual gorilla. For interobserver reliability training, test videos were created using gorillas from a zoo in the United States. Group scans of association patterns were conducted on these videos by all data collectors and compared to the coding of the videos by AL. All observers scored greater than 90% agreement for the videos. Additional in-person training was provided by DK via informal interobserver reliability sessions where DK and the trainee data collector simultaneously observed the gorillas and discussed the observed association patterns.

## Data processing

Nesting and shifting data were each a single data point per day, thus a one/zero association score for each gorilla pair was tabulated per day for both contexts. If a gorilla was not visible at the time of the group scan, they were counted as not visible for the observation. Mixing yard data were collected over three scans. To reduce concerns of autocorrelation between scans conducted in the same day we generated a single one/zero association score for all three scans (i.e. if a pair were associated in at least one of the three scans they were coded as associated for the mixing yard that day). An individual was recorded as not visible if they were not visible for all three scans. An association index for each pair by observation context was calculated using the simple ratio method such that a pair's association was equal to $X/(X+Y_{AB}+Y_A+Y_B)$ where X is the number of observations gorillas A and B were observed associated, $Y_A$ is the number of observations where only gorilla A was observed, $Y_B$ is the number of observations only gorilla B was observed and $Y_{AB}$ was the number of observations both gorillas A and B were observed but were not associated [55].

## Data analysis

All analyses were conducted in SOCPROG Version 2.9 [56] unless stated otherwise and all analyses were conducted independently for each observation context to control for differences in space availability as well as additional factors independent to each context that may have influenced associations. To evaluate group structure we calculated mean association indices and social network measures of *density*, *strength* and *eigenvector centrality*. *Density* is a

reflection of the sum of all dyadic connections out of the total possible based on group size and is used as a measure of group level connections [57]. *Strength* is reflective of the sum weight of all connections an individual has within their network [58]. Thus, *strength* is often referred to as an individual's "gregariousness" as it relates directly to the number of association partners one has within a group. *Eigenvector centrality* is a measure reflective of an individual's connection within a social network based on both their *strength* and the *strength* of the conspecifics they are connected to [58]. Thus, individuals with greater *eigenvector centrality* scores are considered to be more "socially central" as both they and their association partners have greater *strength*. Note the terms *strength* and *eigenvector centrality* are used in subsequent sections where formal data analyses are reported and the terms "gregariousness" and "socially central" are used in their interpretation.

To evaluate how demographic variables and individual relationships affected group structure, we first examined how origin and sex correlated with association indices by creating adjacency matrices for origin and sex. For the origin matrix, if individuals had the same origin (i.e. Goma-Goma, Kinigi-Kinigi) then the pair was scored as a one and if the individuals had different origins (i.e. Goma-Kinigi, Goma-Single, Kinigi-Single, Single-Single) then the pair had a score of zero. For the sex matrix, if individuals were of the same sex (i.e. female-female, male-male) the pair was scored as one and if different sexes (i.e. female-male) then the pair was scored as zero. Dietz' R-test was used to correlate each association matrix (matrix of each pair's association index by context) to the origin and sex matrices to evaluate if these factors influenced association patterns (i.e. were gorillas originating from Goma group more likely to associate with other Goma gorillas than the rest of the group). Dietz' R test is a nonparametric matrix correlation test in which the identities of the gorillas in one matrix were randomly permutated 1,000 times and a correlational value of the original and each random matrix was generated. The true correlation value between the two original matrices was then compared to those generated by the random matrices. If the correlation statistic of the original was greater or less than 97.5% of the random permutations then they are considered significantly correlated ($P \leq 0.05$) [58]. We additionally descriptively compared *strength* and *eigenvector centrality* scores amongst origin and sex groups as inferential statistics were not possible due to small sample sizes amongst groups (e.g. $n_{males}$ = 3). To evaluate the influence of age on group structure we compared individual age to their *strength* and *eigenvector centrality* scores using Spearman's rank correlation test R V.4.1.3 [59].

Preferred relationships were analyzed descriptively, using dyadic association indexes compared to the group mean [e.g. 60] and the top three association indices [e.g. 61] as reference points. Specific tests for preferred relationships exist [e.g. 62], however, they are challenging to conduct in larger groups such as this one as the number of multiple pairwise comparisons requires an impractical adjustment of α values [58] and thus, were avoided here.

To evaluate network stability over time, association networks for the first and last 10 weeks of observations were compared using Dietz R-test within each observation context. To test if association indices were consistent between contexts, we used Dietz R-test to conduct pairwise comparisons of each observation context's association indices (Mixing Yard-Nesting; Mixing Yard-Shifting; Nesting-Shifting). Since this by context analysis involved multiple pairwise comparisons, a Bonferroni correction was applied such that α = 0.016.

To test for the presence of socially central individuals we graphed each individual's *eigenvector centrality* against their reverse *eigenvector centrality* rank (i.e. greatest centrality score was ranked 14) for each observation condition [63, 64]. We then fit both a linear ($y = α + βx$) and power regression ($y = αx^β$) to these data using SPSS V.24 where y is the dependent variable (*eigenvector centrality*), x is the independent variable (rank), α is the y-intercept and β is the slope. The fit of each regression was compared descriptively using the Akaike Information

Criterion (AIC) calculated using the residual sum of squares (RSS) such that AIC = n*[ln(RSS/n)] + 2k, where n is the number of gorillas observed and k is the degrees of freedom. AIC calculated with RSS was used to assess regression fit as the data were normally distributed, as determined by Shapiro-Wilk test (P > 0.05) and visual inspection of QQ plots. A stronger linear regression fit indicates the social network is random or not influenced by specific "hyper-central" individuals [65]. A stronger power regression fit would indicate the social network is influenced by a more "hyper-central" individual(s), described as a scale-free network [66].

To evaluate the presence or absence of subgrouping within the group we calculated a modularity score (Q) developed by Newman [67] for which a group modularity value is calculated based on the observed and expected proportions of clusters (subgroupings) within a population. Modularity values range from 0.0 (random clusters of individuals) to 1.0 (individuals only associated with conspecifics in their cluster), with a value ≥ 0.30 considered an indication of subgrouping in a population [67]. To-date, no work has clarified if a Q value ≥ 0.30 is biologically relevant for gorillas, thus based on the recommendations of Kasper & Voelkl [68] any Q value ≥ 0.30 was followed up with additional analysis to detect if biologically relevant factors were correlated with the clusters. For this follow up analysis a Dietz' R-test was used to determine whether subgroup membership was defined by shared origin or sex.

## Visualization

Association matrices were visualized as network diagrams in Gephi v. 0.10.1 [69]. Placement of nodes within each diagram was based on the Force Atlas algorithm such that more strongly linked nodes were placed closer together than more weakly linked nodes. Node size was proportional to node *strength* and edge weights were proportional to the nodes' association index.

## Results

### Association indices

*Density* was high across all three contexts ($d_{mixing}$ = 1.00; $d_{nesting}$ = 0.98; $d_{shifting}$ = 0.99), with only three pairs not nobserved associating across two contexts (out of 91/context; Fig 1). Despite a high density, mean association indices approached zero across contexts ($\mu_{mixing}$ = 0.05, SD = 0.04; $\mu_{nesting}$ = 0.07, SD = 0.11; $\mu_{shifting}$ = 0.05, SD = 0.05; S1-S3 Tables in S1 File). Shared origin was not correlated with association indices in any of the contexts ($R_{mixing}$ = 0.134, P = 0.208; $R_{nesting}$ = 0.059, P = 0.487; $R_{shifting}$ = 0.151, P = 0.172; Fig 2A). Sex was not correlated with association indices in any of the contexts ($R_{mixing}$ = 0.095, P = 0.51; $R_{nesting}$ = 0.117, P = 0.324; $R_{shifting}$ = 0.116, P = 0.499; Fig 2B).

Association rates were greater than the group mean for 39.6% (36/91), 23.1% (21/91), and 28.6% (26/91) of dyads in the mixing, nesting and shifting contexts, respectively, suggesting there was a level of selectiveness in how the gorillas chose to associate. Adult females AMA and MAP maintained one of the three highest association indices in each of the three observation contexts, with values ranging from 5.29 to 6.93 times greater than the group means. Adult females MUY and NDJ similarly maintained one of the three highest association indices in the mixing and shifting contexts with association values 3.47 and 3.39 times greater than the group means, respectively. The remaining dyads with one of the three highest association indices within an observation context included silverback KIG and adult female KAL (7.07 times greater than nesting mean), subadult female LUL and adult female PIN (8.27 times greater than nesting mean), KIG and MUY (3.47 times greater than mixing yard mean), and AMA and adult female ISA (3.90 times greater than shifting mean). The three males maintained association indices that were approximately equal to group means across contexts except for in the nesting context. Blackbacks LUB and SHA were the fifth most frequently associated dyad

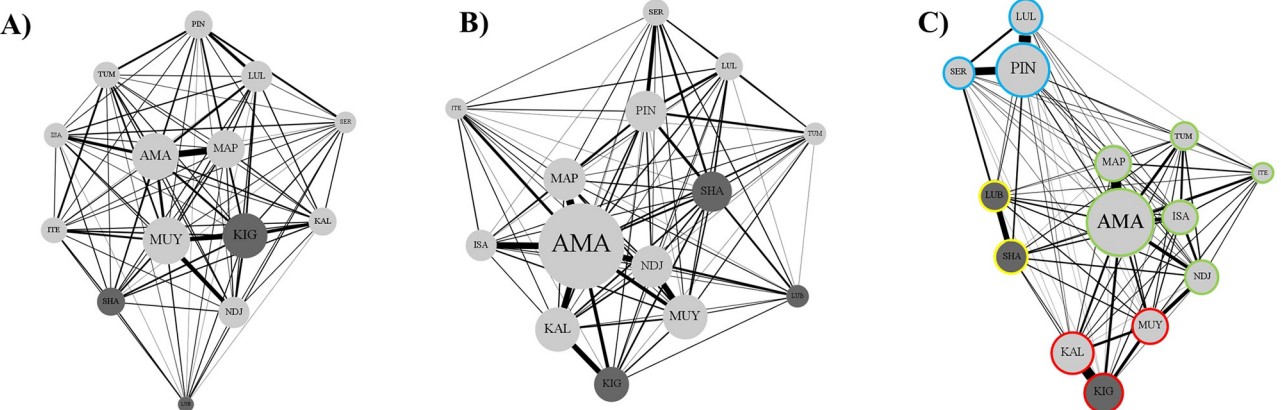

**Fig 1. Social network diagrams by observation context.** (A) mixing yard, (B) shifting and (C) nesting. Dark grey nodes indicates males. Light grey nodes indicates females. For (C), evidence of subgrouping was observed via modularity tests (Q = 0.383), membership in each of the four identified subgroups is noted by node outline color.

in the nesting context with an association index 3.87 times greater than the group mean. In contrast, KIG and LUB were unobserved to be associated in the nesting context (only one of two pairs unobserved in nesting context). The other pair not observed associating in the nesting context was LUL and adult female TUM. Additionally, KIG and ISA were not observed associating in the shifting context.

## Social network statistics

Mean *strength* ranged from 0.64 (SD = 0.19) to 0.97 (SD = 0.30) by context and mean *eigenvector centrality* ranged from 0.25 (SD = 0.08) to 0.26 (SD = 0.07) by context (Table 2). Gorillas originating from Goma had a greater *strength* and e*igenvector centrality* score than gorillas originating from Kinigi or Single in all three observation contexts (Table 3). Female gorillas also had greater *strength* and *eigenvector centrality* scores than male gorillas in all three observation contexts but differences between males and females were relatively small (Table 3). Age was not correlated to either *strength* or *eigenvector centrality* across all three contexts (mixing,

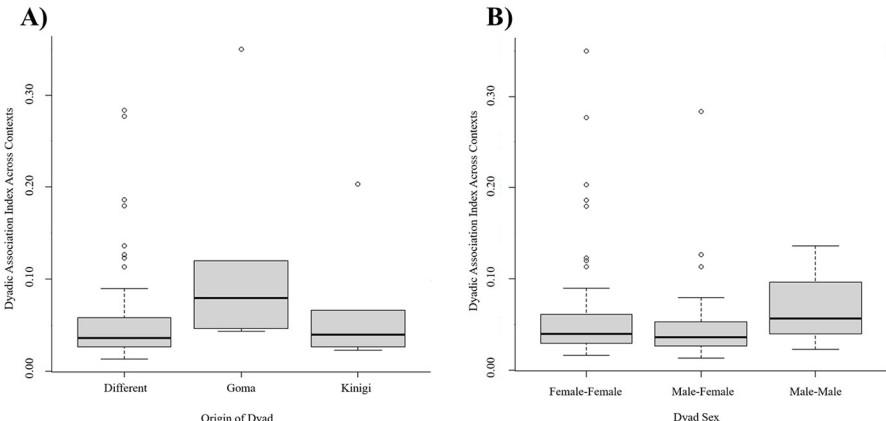

**Fig 2. Mean dyadic association indices by dyad classification.** (A) origin of dyad and (B) sex of dyad.

**Table 2. Strength and eigenvector centrality scores by gorilla and association context.** Bolded values are the three highest values for each measure/context.

| Gorilla | Mixing Yard | | Nesting | | Shifting | | Individual Mean Across Contexts | |
|---|---|---|---|---|---|---|---|---|
| | Strength | Eigenvector Centrality | Strength | Eigenvector Centrality | Strength | Eigenvector Centrality | Strength (SD) | Eigenvector Centrality (SD) |
| AMA | **0.95** | **0.39** | **1.77** | **0.43** | **1.52** | **0.51** | **1.41 (0.34)** | **0.44 (0.05)** |
| ISA | 0.50 | 0.20 | 0.92 | 0.24 | 0.55 | 0.25 | 0.66 (0.19) | 0.23 (0.02) |
| ITE | 0.53 | 0.21 | 0.56 | 0.16 | 0.38 | 0.16 | 0.49 (0.08) | 0.18 (0.02) |
| KAL | 0.57 | 0.24 | **1.05** | 0.28 | **0.79** | **0.32** | 0.80 (0.20) | 0.28 (0.03) |
| KIG | **0.91** | **0.37** | 1.00 | 0.27 | 0.62 | 0.26 | 0.84 (0.16) | 0.30 (0.05) |
| LUB | 0.32 | 0.13 | 0.82 | 0.19 | 0.40 | 0.15 | 0.51 (0.22) | 0.16 (0.02) |
| LUL | 0.63 | 0.25 | 0.92 | 0.29 | 0.49 | 0.16 | 0.68 (0.18) | 0.23 (0.05) |
| MAP | 0.77 | 0.34 | 0.95 | **0.31** | 0.75 | **0.32** | 0.82 (0.09) | **0.32 (0.01)** |
| MUY | **0.96** | **0.38** | 0.94 | 0.24 | **0.79** | 0.30 | **0.90 (0.08)** | **0.31 (0.06)** |
| NDJ | 0.63 | 0.27 | 0.87 | 0.23 | 0.74 | 0.29 | 0.75 (0.10) | 0.26 (0.02) |
| PIN | 0.57 | 0.20 | **1.42** | **0.35** | 0.73 | 0.22 | **0.91 (0.37)** | 0.26 (0.07) |
| SER | 0.44 | 0.16 | 0.80 | 0.23 | 0.47 | 0.14 | 0.57 (0.16) | 0.18 (0.04) |
| SHA | 0.57 | 0.22 | 0.82 | 0.19 | 0.71 | 0.23 | 0.70 (0.10) | 0.21 (0.02) |
| TUM | 0.54 | 0.21 | 0.71 | 0.19 | 0.40 | 0.14 | 0.55 (0.13) | 0.18 (0.03) |
| **Group Mean (SD)** | 0.64 (0.19) | 0.25 (0.08) | 0.97 (0.30) | 0.26 (0.07) | 0.67 (0.29) | 0.25 (0.10) | 0.76 (0.23) | 0.25 (0.07) |

$r_{strength}$ = -0.178, P = 0.542, $r_{eigenvector}$ = -0.245, P = 0.399; nesting, $r_{strength}$ = -0.138, P = 0.638, $r_{eigenvector}$ = -0.048, P = 0.871; shifting, $r_{strength}$ = -0.237, P = 0.414, $r_{eigenvector}$ = -0.287, P = 0.319).

Silverback KIG had one of the three highest *strength* and *eigenvector centrality* scores in mixing yard observations, but was approximately equal to group mean values in the nesting and shifting contexts (Table 2). Blackback SHA was just below the group mean *strength* and

**Table 3. Network analysis measures for gorilla origin and sex by association context.** Bolded values are the highest values for each measure by grouping variable and association context.

| Association Context | Grouping Variable | Subgrouping Variable | Strength (SD) | Eigenvector Centrality (SD) |
|---|---|---|---|---|
| Mixing Yard | Origin | Goma | **0.82 (0.15)** | **0.34 (0.05)** |
| | | Kinigi | 0.52 (0.06) | 0.20 (0.02) |
| | | Single | 0.59 (0.21) | 0.24 (0.08) |
| | Sex | Male | 0.60 (0.29) | 0.24 (0.12) |
| | | Female | **0.64 (0.17)** | **0.26 (0.08)** |
| Nesting | Origin | Goma | **1.15 (0.42)** | **0.31 (0.09)** |
| | | Kinigi | 0.87 (0.38) | 0.23 (0.09) |
| | | Single | 0.91 (0.09) | 0.24 (0.04) |
| | Sex | Male | 0.88 (0.10) | 0.22 (0.05) |
| | | Female | **0.99 (0.33)** | **0.27 (0.08)** |
| Shifting | Origin | Goma | **0.91 (0.41)** | **0.35 (0.11)** |
| | | Kinigi | 0.50 (0.16) | 0.17 (0.04) |
| | | Single | 0.62 (0.16) | 0.24 (0.07) |
| | Sex | Male | 0.58 (0.16) | 0.21 (0.05) |
| | | Female | **0.69 (0.32)** | **0.26 (0.11)** |

*eigenvector centrality* scores across contexts, except for shifting where his *strength* was above the mean. Blackback LUB was below average in both metrics across all contexts and generally maintained the lowest or one of the lowest scores in each context. Adult female AMA had the highest *strength* and *eigenvector centrality* scores across all three contexts. Averaging across contexts, adult females AMA, MUY and PIN had the highest *strength* and adult females AMA, MAP and MUY had the highest *eigenvector centrality* scores within the group.

## Consistency of association indices

In each observation context, the association indices observed in the first and last 10 study weeks were significantly correlated ($R_{nesting}$ = 0.377, P < 0.001; $R_{mixing}$ = 0.236, P = 0.04; $R_{shifting}$ = 0.329, P = 0.004), suggesting association patterns were consistent over time within each context. Pair wise comparisons of association indices between observation contexts found that indices were also significantly correlated ($R_{mixing-nesting}$ = 0.329, P = 0.006; $R_{mixing-shifting}$ = 0.404, P = 0.002; $R_{nesting-shifting}$ = 0.511, P < 0.001), suggesting association indices were similar between observation contexts.

## "Hyper-sociality" and subgroupings

No evidence of a hyper-central individual was observed, as the cumulative distribution of the group's *eigenvector centrality* scores more closely followed a linear distribution than a power function in each context: mixing yard (linear, AIC = -80.413, $R^2$ = 0.927, $F_{1,12}$ = 152.616, P < 0.001; power, AIC = -35.032, $R^2$ = 0.874, $F_{1,12}$ = 83.133, P <0.001; Fig 3A), shifting (linear, AIC = -65.032, $R^2$ = 0.840, $F_{1,12}$ = 63.194, P < 0.001; power, AIC = -23.566, $R^2$ = 0.773, $F_{1,12}$ = 45.392, P <0.001; Fig 3B) and nesting (linear, AIC = -78.543, $R^2$ = 0.889, $F_{1,12}$ = 95.728, P < 0.001; power, AIC = -36.004, $R^2$ = 0.822, $F_{1,12}$ = 55.362, P <0.001; Fig 3C). This suggests the social structure of the group was random and not scale-free.

For both mixing yard (Q = 0.161) and shifting contexts (Q = 0.214), modularity scores were below 0.30, suggesting subgroups were not present. In the nesting context, modularity scores exceeded the expected threshold (Q = 0.383), revealing four clusters of gorillas within the group (Fig 1C). Nesting subgroup membership was not defined by shared origin (R = 0.065, P = 0.461) but membership was significantly defined by sex (R = 0.319, P = 0.032). Three subgroups were single sex ($n_{female}$ = 2, $n_{male}$ = 1) and one was mixed-sex, composed of silverback KIG and two adult females MUY and KAL.

## Discussion

### Association indices

Across contexts the group was highly connected, with densities approaching or equal to one. In contrast, mean association indices approached zero. Thus all gorillas were generally observed associating with one another but they associated infrequently. The high densities may be partially reflective of the sanctuary environment such that the chances of being in proximity to a conspecific increased given they lived in a finite amount of space. The low association indices amongst individuals are similar to those reported between male-female mountain gorilla dyads within 2m in nature [4] suggesting the GRACE gorillas associate with one another similarly to mountain gorillas, their closest relatives. Association patterns of gorillas in zoos vary widely by study making comparisons to our findings challenging [49, 70–72]. This may be reflective of the diversity of group-types, living spaces and management procedures within zoos, and possibly because the reporting of association patterns in zoos are often focused on a specific applied perspective, rather than a broad description of gorilla life in

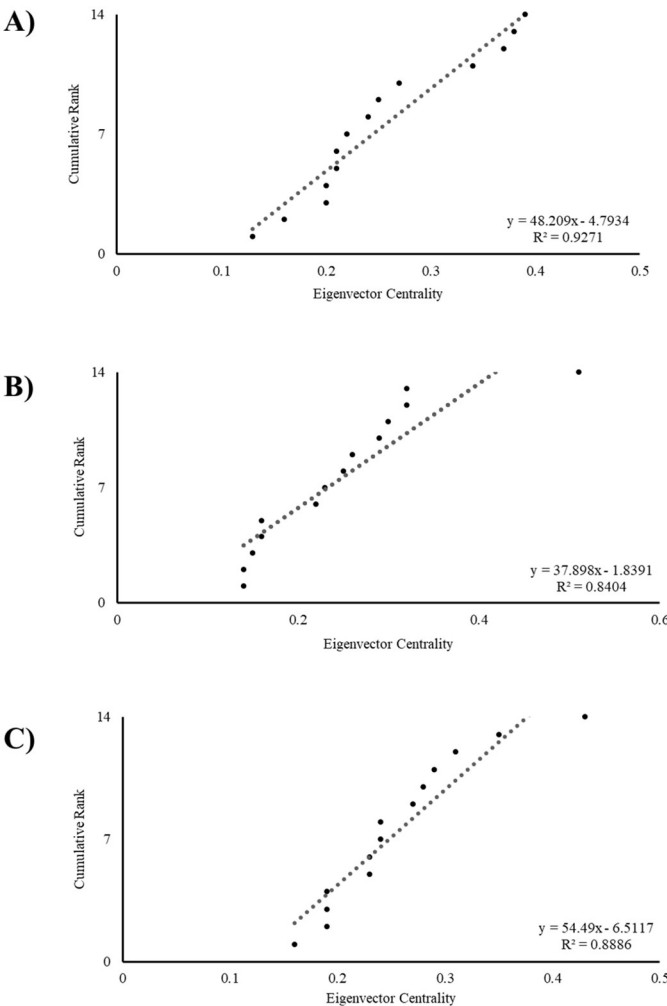

**Fig 3. Cumulative rank and distribution of eigenvector centrality scores by observation context.** (A) mixing yard, (B) shifting and (C) nesting. The dotted line represents the linear regression fit of the data.

general. To this latter point, we hope that zoos begin to place a stronger emphasis on this line of inquiry as it could be used to better understand the sociality of gorillas in both zoos and in-situ sanctuaries.

Despite the overall weak association indices, up to a third of the dyads associated at rates greater than the group mean, suggesting the gorillas did differentiate their relationships. In nature, the strongest association patterns amongst gorillas are observed between mothers and infants [4]. No mother-infant pairs were present in this study, however, the youngest gorilla LUL was introduced to the group when she was approximately 18 months old, six years prior to this study, and was surrogate reared by adult female PIN. Their association indices were above average across contexts, and their nesting association index was the single highest association index of any pair in any context, suggesting they maintained a strong relationship six years following their introduction, likely similar to a naturally occurring mother-offspring relationship. Interestingly, as the youngest individual in the group, LUL did not appear to have a strong relationship with silverback KIG. It could be that because LUL was introduced to the

group before KIG developed into a silverback that the affinity observed between adult males and maturing individuals in mountain gorillas was unobserved here [4].

The majority of dyads associating at rates above the group mean were female-female. In mountain gorillas where female dispersion is not obligate, matrilineal biases amongst female's association patterns have been observed [73]. As familiarity is the presumed underlying proximate mechanism driving kin biased behavior [74], it could be that the GRACE females who have lived together for many years since they were young have developed a familiarity with each other that is similar to what is observed in nature. Data from zoos, where females often live in groups for longer periods of time than they would in nature, similarly suggest that stronger association patterns amongst adult females occur compared to what is observed in nature [29, 49]. Thus, these strong female-female relationships may be an artifact of living in human care. An additional factor of living in human care that may influence these gorillas' behavior is the use of oral contraceptives. In humans, the use of contraceptives have been associated with a wide range of behavioral changes [75], including influences on social relationships [76]. Data from a single zoo-living gorilla group suggests the influence of contraceptives on female gorilla behavior may be limited [71]. Data from other primate taxa are also limited and have reported varying levels of influence on behavior [77]. This makes drawing firm conclusions on their influence within our study challenging, however, these drugs remain a potential confounding variable in making comparisons to populations in nature.

It is a promising finding, from the perspective of informing release efforts, that some of the GRACE gorillas maintain comparatively stronger relationships than others as the social support of a bonded partner may assist with mitigating release stressors. What is unknown is how these relationships vary with time. Over this eight month study relationships were stable, however, recent data from mountain and western lowland gorillas suggests that strong female-female relationships are not maintained beyond two years [29]. In Grauer's gorillas, nearly half of female emigrations occur in groups, suggesting females may rely on social support during these significant life events [78]. However, this finding is from a single population of presumably closely related female gorillas and thus may be driven by group demographics not present in the GRACE group. Zoo studies are limited though monitoring of a single group over 12 years suggested female-female relationships were stable long-term [49]. However, this study did not describe these relationships in terms of strength and thus these relationships may have been stable but not necessarily strong. If social bonds amongst gorillas easily diminish with time than relying on association strength as a metric for selecting release candidates may not be useful as bonds may be insufficient to facilitate social support in the face of release stressors.

## Social network statistics

On average, the silverback KIG was more gregarious and socially central than the group mean, however, this varied by context. In the mixing yard he was the third most gregarious and socially central individual, but was approximately average in the nesting (slightly above average) and shifting contexts (slightly below average). Few studies have evaluated social network statistics in relation to age-sex classes within gorilla groups. Rosenbaum et al. [4] conducted the most relevant social network analysis to-date in mountain gorillas, finding that silverbacks were more socially central than the average group member in single male groups, but dominant silverbacks were less central than the average group member in multi-male groups. This suggests that the GRACE group's silverback maintains a similar social position to that of silverbacks in one-male mountain gorilla groups. One notable difference between mountain gorilla silverbacks and the GRACE silverback is that the social centrality of mountain gorillas is partially supported by their relationships with immature individuals [4]. At GRACE, the

silverback maintained a similarly central position despite the group only containing reproductive aged females and maturing males. It may be that in the absence of immature gorillas adult females associate more with the silverback (or vice versa) at GRACE increasing his social centrality, though more study is needed to fully contextualize this difference.

Adult female AMA was the most gregarious and socially central individual across contexts. More broadly the three group members who were the most gregarious and socially central across all contexts were female. The stereotypic description of gorilla social structure is that the silverback is the most socially central individual, thus the results here and those recently from mountain gorillas [4] are surprising in that males appear to be more or less average in their relationships compared to the entire group (which are predominately female). It could be that the sociality of the females observed here are reflective of the uniqueness of the group (i.e. orphans living in human care), however, more detailed reporting of individual group structures of gorillas in-situ are needed to properly contextualize the structure of the group. It could be that female social roles in gorilla societies are more significant than previously considered. Continued application of social network analysis to gorilla datasets may help elucidate this further, potentially highlighting trends unobserved through more traditional descriptions of association patterns.

## Consistency of association indices

Association indices were significantly correlated across our study period suggesting the group's structure was stable over the eight months of study. However, in relation to gorilla life history, eight months is a relatively short period of time. Evaluating this group's social structure over longer periods of time should be informative. For example, do strong relationships between specific dyads end after two years as they do in nature [29] or does group structure change as the two blackbacks develop into silverbacks? To this latter question, currently the data suggest the group is tolerant to the presence of both the silverback and two maturing males. Interestingly, blackback LUB was the least gregarious and socially central gorilla in the group, while SHA was more median but still below average. Amongst themselves, LUB and SHA were one of the most frequently associated nesting dyads, nesting together at a rate approximately four times the group mean. In contrast, LUB and KIG associated below the group mean, with a complete avoidance in the nesting context, while SHA and KIG associated at approximately the group mean. Multi-male groups are rare in Grauer's gorillas and all-male (bachelor) groups have not been reported [35], however, living in human care may provide opportunities for behavioral flexibility not observed in nature. Longitudinal study of these males should be informative to understanding male-male Grauer's relationships, particularly in understanding if young males form relationships within mixed-sex groups that then extend through maturation or if male-male relationships are generally terminal in Grauer's gorillas.

Association indices were also significantly correlated across observation contexts. Harcourt [54] was the first to highlight the importance of activity on gorilla social structure, specifically highlighting that individual relationships, and ultimately group structure, varied between resting and feeding periods. It is surprising then that observations in which food was provisioned (mixing yard) displayed similar social structures to periods of rest (nesting). Given the generally low levels of association in this group, the observation context may not have a significant effect on group social structure. Alternatively, as activity was not directly monitored in these contexts, but rather inferred, it could be that we did not capture a sufficiently broad range of activity patterns to properly differentiate its potential effects on social structure. It is also surprising that association indices were correlated because of the notable differences in space availability in each respective context. Individuals generally associated in similar ways whether

they were in the multi-acre forest enclosure or night house, suggesting association indices were reflective of individual choices and not space constraints.

### "Hyper-sociality" and subgroupings

Our data suggests no one individual was "hyper-social" or disproportionally socially central compared to the rest of the group (i.e. random distribution rather than scale-free). In continuation of a point made earlier, this finding can be considered surprising given the stereotypical description of gorilla social structure centering around the silverback. Interestingly, Kanngiesser et al. [79] suggested primate networks may not typically form scale-free networks, as few studies have found evidence for their presence [68], though a recent study of mandrills (*Mandrillus sphinx*) did report the presence of a scale-free network [63]. It may be that such networks are rare, specific to particular species and/or contexts or may in fact not be an accurate way to describe primate social groupings. As this type of analysis is applied to more primate taxa, its true value in describing social structure should become clearer. Further application of social network analysis within the genus gorilla will additionally help elucidate how silverback social roles can be described within social groups.

In both the mixing yard and shifting contexts, no evidence of subgroupings were observed. However, subgroupings were detected in the nesting context. In nature, gorilla nest location has been associated with environmental variables [e.g. 80] and the need for protection [e.g. 81], but social influences on nesting have received little focus. In mountain gorillas, silverbacks have been described to nest at the edge of the group [82]. In Grauer's gorillas, two studies have suggested nesting patterns are random [83, 84]. In a detailed nesting study of western lowland gorillas in a zoo, Weiche & Anderson [85] reported that blackbacks typically nested out of view of the silverback, there was an affinity for related gorillas to nest together, and some matrilines avoided nesting in association. Here, nesting was not random and was significantly associated with sex such that three of the four subgroups were single sex (two all female, one all male subgroups). Similarly we also found evidence of silverback avoidance by one of the blackbacks during nesting. In human care, overnight management frequently occurs in a more confined space than day time management. This may change group social dynamics by increasing, by default, proximity amongst individuals and minimizing opportunities for avoidance and/or escape. To this point, nesting association may be a highly informative measure of relationship strength for gorillas in human care as it requires a certain trust associated with being near specific individuals during a relatively vulnerable time (i.e. sleeping). Though the subgroupings here do not appear to be random, it is again worth mentioning that the quantitative threshold for identifying subgroupings here is not based on gorilla sociality itself, but rather broad mathematical predictions [67]. From a biological perspective it is unlikely that the same modularity score applied to two different species has the same interpretive value, thus we echo points raised by Kasper & Voelkl [68] that modularity scores themselves should be interpreted cautiously.

Shared experiences are a significant contributor to the formation of social bonds [for review see 86] and thus it could be expected that the gorillas who lived together prior to arriving at GRACE may have maintained preferential relationships with each other based on their prior shared experiences. However, shared origin was not associated with any larger preferential relationships or subgroupings in this group.

In contrast, two separate zoo living chimpanzee groups were introduced and managed as a single group for a year, though social network analysis revealed that they maintained two distinct subgroupings based on their original group origin [87]. Unlike chimpanzees, gorilla dispersal patterns [24] and general absence of territoriality [88, 89 but see 90] may make them less

likely to maintain subgroupings compared to chimpanzees. It may also be that prior origin biases no longer affect individual behavior after certain periods of time, in contrast to the chimpanzee study that was done after one year, the original GRACE gorilla introductions occurred a decade prior to this study. Furthermore, research from nature has reported female gorillas do not maintain preferential relationships beyond two years [29], suggesting origin biases may not be a strong determinant of gorilla social preferences. Though it is worth noting that the most frequently associated pair (adult females AMA and MAP) maintained a shared origin.

## Conclusions

This is the first study to-date to utilize social network analysis to describe the social structure of a Grauer's gorilla group. The data collected here are first and foremost used to inform on the care of these individual gorillas, the only Grauer's gorillas living in human care in the world. The methods used in this study are easily scalable to any facility and because they are collected by the husbandry team a strong buy-in on the use of the data to guide day-to-day care and management decisions was developed. We hope that similar monitoring becomes more common in in-situ primate sanctuaries, ultimately contributing to individual care and welfare and a broader understanding of primate behavior. We additionally hope that these data will be used to support future release efforts. To our knowledge such an approach has only been utilized with lions [91], though it likely has significant application to a variety of species, particularly primates whose survival and wellbeing in any context is heavily influenced by their social relationships. Lastly, in the absence of detailed information from populations in nature this study provides provisional, but important, insights into the social structure of Grauer's gorillas. Though not a replacement or equal in value to findings from in-situ field research, this study adds to a growing body of informative research conducted within in-situ primate sanctuaries [52, 53].

## Supporting information

**S1 Dataset. GRACE_Association_Data_Deposit.** This csv file contains the association data analyzed in the main text.
(CSV)

**S1 File. Supplementary tables.** This word document contains three tables containing the association index for each pair by observation context (S1, mixing; S2, shifting; S3, nesting).
(DOCX)

## Acknowledgments

We are grateful to the entire GRACE team for their support of this project. In addition, GRACE is supported by a team of advisors who provided constructive feedback on the initiation of the behavior monitoring program from which the data here were generated. We are also thankful to Jake Funkhouser for providing feedback on the use of the statistical software SOCPROG.

## Author Contributions

**Conceptualization:** Austin Leeds, Dalmas Kakule, Jackson K. Mbeke, Katie Fawcett.

**Data curation:** Laura Stalter.

**Formal analysis:** Austin Leeds.

**Investigation:** Dalmas Kakule.

**Methodology:** Austin Leeds, Dalmas Kakule, Laura Stalter, Jackson K. Mbeke, Katie Fawcett.

**Visualization:** Austin Leeds.

**Writing – original draft:** Austin Leeds, Laura Stalter.

**Writing – review & editing:** Austin Leeds, Dalmas Kakule, Laura Stalter, Jackson K. Mbeke, Katie Fawcett.

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
