## [Decision Letter · Decision Letter 0]

24 Sep 2023

PONE-D-23-27873Group structure and individual relationships of sanctuary-living Grauer’s gorillas (Gorilla beringei graueri)PLOS ONE

Dear Dr. Leeds,

Thank you for submitting your manuscript to PLOS ONE. After careful consideration, we feel that it has merit but does not fully meet PLOS ONE’s publication criteria as it currently stands. Therefore, we invite you to submit a revised version of the manuscript that addresses the points raised during the review process.

We look forward to receiving your revised manuscript.

Kind regards,

Tomoyoshi Komiyama, Ph.D

Academic Editor

PLOS ONE

Journal Requirements:

Additional Editor Comments:

Dear Authors,

You investigated the stability of social relationships over an eight month observation period and similarities in social structure of the well-studied mountain gorilla subspecies.

Your results suggest that females may be more gregarious and socially central than what is generally considered for gorillas.

You suggested that in addition to contributing to understanding the basic biology of Grauer’s gorillas, the methodology of your study can provide managers of gorillas and other primates a framework to monitor the social dynamics of primate groups in their care.

I have some suggestions for your manuscript that you might consider.

Additionally, the conclusions section was difficult to understand, so please revise it to be more clear and concise.

I think these attached comments will be very helpful in the revision of your manuscript.

Thank you for your time.

Regards

Tomoyoshi Komiyama, Ph.D

Reviewers' comments:

Reviewer's Responses to Questions

**Comments to the Author**

1. Is the manuscript technically sound, and do the data support the conclusions?

Reviewer #1: Yes

Reviewer #2: Yes

2. Has the statistical analysis been performed appropriately and rigorously? 

Reviewer #1: Yes

Reviewer #2: Yes

3. Have the authors made all data underlying the findings in their manuscript fully available?

Reviewer #1: No

Reviewer #2: Yes

4. Is the manuscript presented in an intelligible fashion and written in standard English?

Reviewer #1: Yes

Reviewer #2: Yes

5. Review Comments to the Author

Reviewer #1: Review for Group structure and individual relationships of sanctuary-living Grauer’s gorillas (Gorilla beringei graueri)

Before anything, I will acknowledge that I have limited experience with research on captive animals, so some of my suggestions are just that: suggestions. But I do have a few concerns that go beyond the suggestion part, and to which I would prefer to receive a response (or see edits/changes).

Overall, I think that there are some minor things to adjust and then this can be a good contribution. But, following the authors own words (Discussion), I 'do' think the whole emphasis of the paper should lie a bit more on the application for captive management part, and a bit less on the real-world ecological consequences.

To address that point by point:

First, I am wondering if the whole phrasing of the Introduction, and the general premise for your ‘story’, should not be flipped. Right now, the focus lies heavily on the insights on nature of Grauer’s gorilla social networks themselves (that we can extrapolate from this captive population to the ‘wild’ etc.) and less on the captivity/management part. I would think that, like other studies on zoo/captive populations, this study mainly has value as a work to inform the management of captive Grauer’s’ and provides only tentative insights in the natural history of the ‘wild’ species (more like suggestions for further research, areas where these behaviors differ or confirm what is known from populations of other subspecies). In short: I think you might want to do a little reframing: focus on the applied benefits foremost and only gently make inferences for the larger ecological/evolutionary relevance of your study. EDIT (after reading on): basically like you propose in lines 118-120, focus on paper on objectives 1 and 2 and only briefly touch upon what this could mean for objective 3.

Second, it strikes me that in the paragraphs (Intro) on relationships you make no reference to the importance/our understanding of kinship/kin-specific relationships. That seems to be an integral part of the formation of intragroup associations and I’d think it needs to be mentioned more explicitly (early on in the Intro). See e.g., Morrison, R. E., Groenenberg, M., Breuer, T., Manguette, M. L., & Walsh, P. D. (2019). Hierarchical social modularity in gorillas. Proceedings of the Royal Society B, 286(1906), 20190681.

Third, I’d like to admit that, though I regularly construct statistical models of various types, I am not entirely versed in some of the stats you used (Dietz, modularity score, etc.) and cannot comment on them fully. I do have some smaller remarks that I will explain in the line-by-line comments below.

Other general comments:

-This is a subjective style choice, but I personally think that an active voice would improve the flow of the manuscript. As far as I’m concerned, you can take or leave that suggestion, but I really think it could be beneficial to the overall readability. Again, no need to change it if you don’t want it!

Line-by-line:

29: maybe just to avoid confusion: ‘animals living under/in human care’?

32: officially, I think it’s ‘of the Congo’

32: just a reminder that the Abstract should stand alone, and that the reader does not, at this point, yet know what ‘the GRACE group’ is (are the gorillas in the sanctuary housed together in one group? What does that group look like? Are we talking about 5 gorillas or 20?).

I see some of this throughout the abstract, just make sure that every sentence and every piece of information can only be clearly interpreted in one single way.

This sentence is also a perfect example of how an active voice could make it easier to read: “we monitored…..over the course of eight months….”

32-37: we also don’t yet know what these contexts are, which makes it a bit hard to follow.

40-43: Well, yes, but it is worth re-emphasizing that this is not a natural situation. Social network analyses on mountain gorillas might be more representative of naturally occurring dynamics than a social network found among a captive group. So, I would bring this with more caution.

41: try to avoid long sentences (e.g., instead of continuing with “…, however,” just split it into two sentences). Especially given the fact that you use a passive voice, it becomes a bit lengthy here and there.

56: is this complete (or even the most important ones)? What about demographic factors and kinship? Or do you see that as drivers ‘within’ fitness outcomes? I think it might need to be mentioned explicitly?

59: reference maybe?

67: as you rightly say in the next sentence, this is not the case for mountain gorillas, so isn’t this a bit of an odd statement? Maybe add some modifier such as ‘predominantly’ etc.?

69: I’m not really clear on the ‘by site/year’ part, what does that refer to?

78: be careful with words such as ‘prefer’ (they tend to do this, yes)

90: “of the Congo”

95-97: seems to bit of an odd juxtaposition to me. Sure, they are affected by disturbances, but its unlikely that the response of Grauer’s to human disturbances differs much from that of the other gorilla subspecies.

98: I would say nearly never but ok

99: population rate?

100: worth mentioning though that this was restricted to observations of one small population (4 groups) (of closely related gorillas).

104-106: you do have a disclaimer (“more studies needed”) but even then, I find this a rather tentative hypothesis based on very little data. I don’t think you can say this at this point. Also, “these findings”, you mean those findings that “females with infants more time resting together” or all the findings in the previous sentences?

120-123: I think that it has to be clear early on that these gorillas are not (necessarily) kin/related, unlike a group in the wild (or even most individuals in zoo groups)

142: from wild populations (after all, even captive organisms are natural )

155: 3.11? you mean 3 males and 11 females for a total of 14? I don’t know if this short-hand description works for the broad audience of this journal; would suggest to write it out.

247: Spearman test is base R, no need to call the ‘program’ Rcdmr, just state “…test using R [54]”.

259: Bonferroni

262-270: I think there are some details missing in this paragraph. What did the linear regression look like, e.g., in terms of data family? And the power regression? Power of two I assume …? Also when you say “regression fit statistics including…” then I’m thinking, which other ones?

266: is this Gaussian definition of the AIC the appropriate one? Please ignore my ignorance but it’s not the general definition (as used in e.g., Burnham, K. P.; Anderson, D. R. (2002), Model Selection and Multimodel Inference: A practical information-theoretic approach (2nd ed.), Springer-Verlag.) but an adapted one, and I’m wondering why. Also, might AICc (correction for small sample size) not be more appropriate?

304: just a suggestion but does this all need to written out or can it be put into a table for readability?

392: again, I think the kinship/relatedness part is not addressed enough here.

Reviewer #2: General comments to the authors:

This study provides the first examination of Grauer’s gorilla social behavior in a sanctuary or captive setting. It provides a relatively straightforward and easily scalable social network approach to comparing behavioral association data collected on captive animals in different contexts. The authors provide a clear and succinct background explaining the context and overall aims for the study. They do a great job in detailing the analytical framework that was used. Overall, the paper is well-written, concise, and has important implications for sanctuary managers regarding the maintenance of long-term social associations among dyads of captive gorillas. I have only a few minor questions and suggestions for the authors and have provided detailed in-line comments below.

Specific comments to the authors:

Line 155: Typo. Please clarify group size.

Line 162: Please indicate if females are on birth control. This is important biological information that may influence inter-individual associations in a sanctuary context. Also, please indicate if genetic data is available for these individuals.

Line 172: Please be specific about what these “provided dietary items” were, whether gorillas were allowed to free forage on growing plants within the forest enclosure, and the spatial distribution of food items. If provisioned foods were clumped within the mixing yard or if gorillas were consuming plants in the forest enclosure that were spatially clumped, this may have affected feeding associations.

Lines 178 – 190: It would be helpful if the authors justified why they chose an association threshold of 1 m. Also, was activity of gorillas noted during observations? (i.e., whether feeding, resting, or other?). If not, state this here in methods. Furthermore, the scan sampling protocol suggests more scan data were collected in the mixing yard context (3 scans per observation period) vs. in the nesting or forest enclosure setting (1 scan per observation period). This may affect interpretations of network correlation analyses that were described in Lines 255-260. How did the authors account for temporal autocorrelation for scans collected in the mixing yard context?

Lines 190 – 193: The authors make an interesting observation that the gorillas tended to congregate towards the shift door prior to their recall. How did the authors account for spatial limitations across the three contexts? It is possible that the smaller night houses might have limited inter-individual spacing compared to the outdoor foraging contexts. Thus, sleeping associations may not have been an indication of preference but rather a spatial constraint. It would help to justify this choice. Also, it appears as if the authors do not distinguish between scans that were collected in one forest enclosure over the other. The authors note that the two forests enclosures are different sizes; are there also important differences in how plants are clumped in these two spaces? Some justification on why it was appropriate to lump these observations into one category would be helpful.

Line 263: Clarify how cumulative eigenvector centrality was calculated for each gorilla (i.e., did this include observations from all three contexts).

Line 276: The authors consider a modularity score greater than 0.30 to indicate subgrouping as suggested by Whitehead, but is there any reason to believe this threshold is appropriate for gorillas? Is there evidence (for instance from mountain gorillas) that this low modularity score predicts subgroup membership? It would be useful if the authors could provide some additional justification here and whether this value is biologically relevant for gorillas.

Line 284: In Figure 1 it appears as if the size of these nodes is quite similar. Does this indicate that the sum of individual associations was identical for every gorilla? This does not appear to be the case based on edge weights. This would need to be clarified or the difference in node size needs to be made more apparent.

Line 304: It is a bit unnecessary to list out the specific numbers for dyads here as these are arbitrary based on how the order of dyads were listed in the association matrix. Would be better to rephrase to “three gorilla dyads had an association rate greater…”

Lines 398 – 400: It is intriguing that the females in this group appear to form stronger associations with each other than those observed in the wild. Is there any indication from other captive primate populations that birth control implants (if used) have any effect on female-female interactions? This could be another reason for the observed strength of association among the GRACE females. Without having pre- and post-implant behavioral data, it would be hard to rule this out.

Lines 423 – 428: In addition to the fact that these individuals are in captivity, it is also notable that all the females within this group were at or near the age of reproduction for this species (as noted in Table 1) during the observation period in 2022. In the wild, the high centrality of adult male gorillas is often a function of the close affiliative associations maintained with weaned, younger offspring. This distinction in group age structure cannot be ignored and should be noted here in the discussion.

Conclusions: The authors note in Lines 114 and 118 that the sanctuary’s ultimate goal will be to release gorillas into the wild. It would be beneficial for the authors to indicate (either in the discussion or conclusion) how the observations in this study on dyad-specific associations might influence management decisions. For instance, how might long-term affiliations among specific dyads affect their outcomes if released into the wild?

6. PLOS authors have the option to publish the peer review history of their article (what does this mean?). If published, this will include your full peer review and any attached files.

Reviewer #1: No

Reviewer #2: No

---

## [Author Response · Author response to Decision Letter 0]

7 Nov 2023

Please see the attached document "Response to Reviewers" for full/detailed responses to each edit/comment/question.

---

## [Decision Letter · Decision Letter 1]

27 Nov 2023

Group structure and individual relationships of sanctuary-living Grauer’s gorillas (Gorilla beringei graueri)

PONE-D-23-27873R1

Dear Dr. Leeds,

We’re pleased to inform you that your manuscript has been judged scientifically suitable for publication and will be formally accepted for publication once it meets all outstanding technical requirements.

Kind regards,

Tomoyoshi Komiyama, Ph.D

Academic Editor

PLOS ONE

Additional Editor Comments (optional):

Dear authors,

Thank you for submitting your revised manuscript.

It was much easier to understand than the original manuscript.

I am satisfied with the responses and the edits, so I am happy to accept your study.

You have satisfactorily addressed the comments from the two reviewers.

Therefore, I have no further suggestions.

I believe this manuscript will satiate the reader's interest.

Tomoyoshi Komiyama

Reviewers' comments:

Reviewer's Responses to Questions

**Comments to the Author**

1. If the authors have adequately addressed your comments raised in a previous round of review and you feel that this manuscript is now acceptable for publication, you may indicate that here to bypass the “Comments to the Author” section, enter your conflict of interest statement in the “Confidential to Editor” section, and submit your "Accept" recommendation.

Reviewer #1: All comments have been addressed

Reviewer #2: All comments have been addressed

2. Is the manuscript technically sound, and do the data support the conclusions?

Reviewer #1: Yes

Reviewer #2: Yes

3. Has the statistical analysis been performed appropriately and rigorously? 

Reviewer #1: Yes

Reviewer #2: Yes

4. Have the authors made all data underlying the findings in their manuscript fully available?

Reviewer #1: Yes

Reviewer #2: Yes

5. Is the manuscript presented in an intelligible fashion and written in standard English?

Reviewer #1: Yes

Reviewer #2: Yes

6. Review Comments to the Author

Reviewer #1: I think that all the comments have been addressed appropriately and effectively, good job. Perhaps you might want to check one last time, before proofs are finalized, whether you've used your active voice consistently (e.g., line 512?). One final read over should suffice.

Reviewer #2: Thank you for your detailed comments to the concerns raised during the review process. All initial concerns have been adequately addressed. In response to the authors' question clarifying what was meant by "genetic data", we believe the issue has been addressed in a comment made by Reviewer 1 regarding kin structure of the group (i.e. that gorillas are unrelated). In response to the authors' response on the modularity score, thank you for this detailed response, we believe this explanation to be satisfactory.

7. PLOS authors have the option to publish the peer review history of their article (what does this mean?). If published, this will include your full peer review and any attached files.

Reviewer #1: No

Reviewer #2: No

---

## [Editor Report · Acceptance letter]

29 Nov 2023

PONE-D-23-27873R1 

Group structure and individual relationships of sanctuary-living Grauer’s gorillas (*Gorilla beringei graueri*) 

Dear Dr. Leeds:

I'm pleased to inform you that your manuscript has been deemed suitable for publication in PLOS ONE. Congratulations! Your manuscript is now with our production department. 

Kind regards, 

on behalf of

Dr. Tomoyoshi Komiyama 

Academic Editor

PLOS ONE